# Being a Group Fitness Instructor during the COVID-19 Crisis: Navigating Professional Identity, Social Distancing, and Community

**Karin Andersson** [1,*] and **Jesper Andreasson** [2]

1   Department of Sport Sciences, Malmö University, Nordenskjöldsgatan 10, E436 Orkanen, Sweden
2   Department of Sport Science, Linnaeus University, 39182 Kalmar, Sweden; jesper.andreasson@lnu.se
*   Correspondence: karin.jemima.andersson@mau.se

**Abstract:** Research question and purpose: Les Mills is a New Zealand-based fitness distributor with a community consisting of approximately 140.000 instructors worldwide who teach standardized work-out routines. This paper aims to explore how the COVID-19 pandemic and related measurements, such as social distancing affect the everyday lives and professions of Les Mills International (LMI) group fitness instructors. The aim was met with the following research questions: RQ1: How are social distancing and social connectedness understood, and how do they condition LMI instructors' understanding of their profession? RQ2: What do LMI instructors think about the #LesMillsUnited campaign to maintain a strong trainer community in the midst of the pandemic? RQ3: How do LMI instructors think that group fitness will change long term due to social distancing? Research methods: Using qualitative measures and a case-study-based approach, data were gathered through interviews with LMI-certified group fitness instructors. Seven semi-structured focused group discussions with fifteen group fitness instructors from different countries were conducted and audio recorded. The first round of virtual discussions took place in April 2020, and the follow-up talks in September 2020. A thematic analysis was employed to analyze the material. Results and findings: According to the participants, online classes as a means of upholding group fitness in times of social distancing is an insufficient substitute to face-to-face instructing, lacking social connectedness that is normally maintained through successful rituals or social scripts. Navigating "instructorhood" during the pandemic includes emotional labor where not only relationships to clients are challenged, but instructors also experience societal pressure to reinvent themselves as instructors. Implications: With no way of telling how long social distancing needs to be practiced, the group fitness industry is facing unprecedented challenges. Making sense of the group fitness profession currently preoccupies instructors who may now have to redefine to themselves how they can teach, and who for.

**Keywords:** COVID-19; fitness instructor; Les Mills; healthism; gym and fitness culture; professional identity





## 1. Introduction

The year 2020 will inevitably be remembered for one of the worst global health crises in modern times, the COVID-19 pandemic. Although the world has experienced pandemics before, such as severe acute respiratory syndrome (SARS) (2003), swine flu (2009), and Ebola (2014), scholars foreshadow that COVID-19 will have a much more severe impact on society (Malcolm and Velija 2020, p. 29). Evidently, the trust in modern medicine and technology has provided a feeling of security that has been put to the test. Strategies for defeating COVID-19, such as lockdowns, curfews, and social distancing, were implemented and referred to as "the new normality". Disregarding the severity of governmental measurements taken, or in which country one resides, hardly anyone could claim not to have been affected at all, and some reports even point toward a standstill in globalization (Evans et al. 2020, p. 85; Krieger et al. 2021). In addition, the consequences

of governmental restrictions have shown which commercial and non-profit industries are self-reliant—and which are not.

The health, fitness, and sport industry, which prior to the pandemic was worth USD 4 trillion, rapidly found itself struggling, following cancelled tournaments and games (Tiller 2020). In fact, due to restrictions or voluntary actions, two thirds of the world's health and fitness facilities either took temporary breaks or shut down their businesses completely (Health Club Management 2020). On a global scale, various sporting mega-events were cancelled, e.g., the Summer Olympic Games in Tokyo (Krieger et al. 2021). Still, alternative strategies were implemented, such as the so-called NBA bubble in Florida, US, in which the basketball season continued under strict isolation and surveillance (Krieger et al. 2021). Furthermore, the commercial arena of gym and fitness has suffered comparable losses; for example, in countries such as Israel, Jordan, and Australia, gyms and fitness centers were closed down completely for months during full or partial lockdowns. In other countries, such as Finland, Sweden, and Austria, decreasing numbers of gym-goers and cancelled memberships illustrate the impact of social distancing (IHRSA 2021).

Les Mills International (LMI) is, based on its revenue, the worlds' largest provider of pre-choreographed workouts (Parviainen 2011). Prior to the COVID-19 pandemic, LMI's exercise routines were taught in more than one hundred countries by some 140,000 instructors in 20,000 clubs worldwide (LMI 2020). Their different workout programs, such as Bodypump™, Bodyattack™, Sh´bam™, Sprint™, and others, have all been designed for different "feels", making exercise a matter of social lifestyle events (Andreasson and Johansson 2015). Gathered under the umbrella term of "exertainment", a portmanteau of "exercise" and "entertainment", these programs have, since the turn of the century, made a lasting mark on gym and fitness culture, reaching millions of fitness enthusiasts on a weekly basis. Therefore, they can also be considered global producers of body ideals, health, and lifestyles. The underlying pedagogy of their fitness programs aims to provide a social situation in which participants, guided by licensed group fitness instructors, reap strong emotional and social benefits while exercising and gaining physiological benefits (Andreasson and Johansson 2015). Consequently, as LMI programs are to be understood as social events—essentially "working out together"—group fitness instructors (and clients) may face severe consequences due to the COVID-19 crisis. Although LMI has continued to offer workout routines and group fitness programs through different online services, they estimate substantial losses, since 31% of their partnering clubs have switched to online classes, while many others are cutting costs or going bankrupt (LMI 2020).

In order to gain a deeper understanding of how a global crisis such as COVID-19 may affect professionals of another global phenomenon; fitness, this paper examines the experiences of LMI group fitness instructors and how the fitness landscape is developing and transitioning in the midst of the COVID-19 pandemic. Building on qualitative data gathered through focused online group discussions and a case study-based approach to the research, the paper aims to explore how the COVID-19 pandemic and social distancing affect the everyday lives and careers of group fitness instructors. Participating instructors completed LMI training in which they were taught how to teach standardized classes with the goal of turning participants into regulars. They were also encouraged to socialize with other trainers in the LMI community. These endeavors or rituals (Collins 2004) largely rely on joint workouts and social time spent before and after class at the gym. Generally speaking, the pandemic has interrupted the learned and familiar behaviors involved in being an "authentic" instructor, admittedly simplified, dividing the trainer community into camps of individuals who wish to "go on as normal" and others who wish to practice social distancing. The aim of the research was met by means of the following research questions:

RQ1: How are social distancing and social connectedness understood, and how do they condition LMI instructors' understanding of their profession?

RQ2: What do LMI instructors think about the #LesMillsUnited campaign to maintain a strong trainer community in the midst of the pandemic?

RQ3: How do LMI instructors think that group fitness will change long term due to social distancing?

The paper begins with a general background, firstly on research done on physical activity during a crisis, and secondly on fitness professionals and the LMI company. The following sections present the analytical framework for the paper and the research design for the study, followed by the findings in which the aforementioned research questions are addressed sequentially. Lastly, a discussion summarizes and elaborates on the results of the paper.

## 2. Background

### 2.1. Health and Physical Activity during a Crisis

While COVID-19 began to roam the globe in 2020, it has been argued that another pandemic was being fought simultaneously—that of obesity. Allegedly caused by lacking physical activity[1] (physical inactivity) and other seemingly risky lifestyle choices, such as smoking and fast food consumption, obesity was labeled a pandemic by the World Health Organization (WHO) in 2012. Obesity leads to 3,200,000 deaths yearly (Hall et al. 2020, p. 1), which, in comparison, is tenfold to COVID-19-related deaths (Papaioannou et al. 2020, p. 414). Both COVID-19 and obesity could consequently be considered immediate societal issues where individuals might personally experience severe consequences unless they act, which could explain drastic measures. Indeed, some of the immediate concerns following COVID-19 lockdowns and curfews were the reduction in physical activity that would potentially increase. Chen et al. (2020), for example, have suggested that:

> It is likely that prolonged home stay may lead to increased inactivity, such as spending excessive amounts of time sitting, reclining, or lying down for screening activities. (p. 103)

COVID-19 could, thus, feed an already existing global issue. However, recent studies rather convey that adults who were active before COVID-19 maintain their activity levels (Norberg et al. 2021; Scheerder et al. 2020; Brand et al. 2020). Nevertheless, some scholars are now fearing a third pandemic related to mental illnesses, as a direct result of quarantine habitus and social distancing (Faulkner et al. 2021, p. 322). Although some countries, such as Sweden and Finland, has kept gyms open, people belonging to risk groups, e.g., individuals with respiratory diseases, cancer survivors, or people with cardiovascular conditions, are likely to avoid these spaces, although they might be in particular need of physical exercise (Papaioannou et al. 2020, p. 416; see also IHRSA 2021).

Although the changes in exercise patterns and levels of physical activity during and post-corona are still to be researched, nationally as well as internationally (Krieger et al. 2021; Pedersen et al. 2020), studies have been conducted on exercise habits in relation to other types of disasters. For example, in 2011, Eastern Japan suffered a severe tsunami and hurricane, after which researchers reported that a decrease in physical activity could be observed as long as three years after the incident (Okazaki et al. 2015, p. 722). Hall et al. (2020) predict that COVID-19 will have a similar effect. Correspondingly, it has been suggested by Chen et al. (2020, p. 104) that "exercising at home using various safe, simple, and easily implementable exercises is well suited to avoid the coronavirus and maintain fitness levels". Moderate physical load has also proven to have positive mental effects to counteract:

> . . . stressor factors such as longer quarantine duration, infection fears, frustration, boredom, inadequate supplies, inadequate information, financial loss, and stigma. (Jiménez-Pavón et al. 2020, p. 1)

---

[1] We define PA as follows: "any bodily movement produced by skeletal muscles that require energy expenditure, whereas exercise is a subcategory of PA that is planned, structured and repetitive, and aims to improve or maintain one or more components of physical fitness" (Faulkner et al. 2021, p. 321).

Concurrently, health and exercise seem to represent what Rushton and Williams (2012) refer to as a paradigm of "security", meaning that the fear of contracting COVID-19 drives individuals to pursue seemingly healthy lifestyles. Viewed from a wider perspective, fitness practices and the pursuit of health have already been studied under the umbrella term "healthism" (Crawford 1980), which is an ideology interwoven with neoliberal consumerism—an urge to perfect one's body by means of meticulous training, nutrition, and accompanying purchasable products—in the belief that one's outer appearance reflects inner moral values (Sassatelli 2000; Markula 2019; Dworkin and Wachs 2009; Hilgers 2013; Mansfield 2011; Nash 2016; Tolvhed and Hakola 2018; Howell and Ingham 2001). Wendy Brown states that "a neoliberal rationality, while foregrounding the market, is not only or even primarily focused on the economy; rather it involves extending and disseminating market values to all institutions and social action" (2012, p. 7). From this perspective, healthism is understood as a governmentality that "casts people as human capital who must constantly tend to their own present and future value" (Brown 2012, p. 54). This logic becomes especially evident within this study as instructors underline their individual responsibilities of staying fit and entrepreneurial despite the crisis—preparing for a post-corona. In the words of an informant, "I decided that this was my time to do the outmost to better myself as an instructor." Since fitness instructors are often portrayed as "pillars of health" (Safai 2017), one can assume that many fitness professionals were already practicing healthy lifestyles before the COVID-19 pandemic; nevertheless, the pandemic could increase the pressure of upkeeping such habits (Andersson et al. 2021b, p. 175).

### 2.2. Fitness Professionalism—A Culture in Transition

Research on the occupation of group fitness leaders has a relatively short history. However, since the turn of the century, and as a result of the global expansion and growth of gym and fitness culture (Sassatelli 2000; Andreasson and Johansson 2014), scholars have increasingly directed their attention toward fitness professionalism, and how the occupation of a fitness professional can be understood in relation to such concepts as education, ethical boundaries, selection processes, and occupational legitimacy (Felstead et al. 2007; Tinning 2010; George 2008). Group fitness activities, and "instructorhood", have also been studied in terms of standardization and McDonaldization (Andreasson and Johansson 2016). Parviainen (2011) suggests that the concept of standardization seems to capture the changing educational and other conditions in the labor market for fitness professionals. To become a certified LMI group fitness instructor, trainees take part in a two-day workshop in which they learn about movement patterns and the shared values of the community. Subsequently, in order to become officially certified, trainees are given a time frame to send in a video of themselves teaching the pre-choreographed routine they have learned. The pre-choreographed routine is to be understood as incorporating relatively easy movement patterns (choreography), combined with music in more complex sequences, much like the principles of scientific management developed by Frederick Taylor (Parviainen 2011). Importantly, apart from producing standardized classes and movements, LMI highly values their community, to which they refer as a global fitness family (LMI 2020). They point out that in choosing music and preparing classes, they take cultural differences and ways of life seriously. Their self-proclaimed goal is to spread fitness through social togetherness—to create a fitter planet (LMI 2020).

Since the outbreak of the pandemic, LMI (as many other fitness providers) has engaged in offering fitness on-demand through streaming services. Since the teaching materials used were already being delivered in digital form prior to COVID-19, moving into virtual fitness was not a substantial step for the company, but rather an acceleration of an already ongoing process. Additionally, since physical exercise is an important aspect of fending off disease and maintaining a strong immune system, "physical exercise emerges as a cornerstone, as a preventive measure to improve host defense against influenza viral infection and other metabolic diseases" (Luzi and Radaelli 2020, p. 4); LMI simply needed to re-focus its

marketing to on-demand services during lockdown.[2] However, to the practitioners, it was a sudden rupture of professional habits.

Aware of the precarious situation in gyms, LMI launched a campaign called #LesMillsUnited in which instructors are encouraged to film short clips of themselves exercising and upload them to their social media accounts with the corresponding hashtag. The campaign can be seen as an example of a global enterprise (re-)navigating social interaction during a crisis, but also as an appeal to exercise, similar to the campaign by Sport England, "join the movement"—#Stayinworkout (Malcolm and Velija 2020, p. 26). In this paper, we direct attention toward the LMI campaign as well as other ways in which individual instructors navigate a new fitness landscape and enact social distancing in a profession based on social connectivity and a culture of togetherness.

### 3. Analytical Framework

Taking a social constructionist approach, we consider the ways in which fitness professionals navigate their professional identities as instructors, and how they "do" instructorhood, in a time of social distancing during a global pandemic. For our discussion, we define professionalism according to a definition by Charles Goodwin (1994), who argues that professionalism is a socially constructed and situated way of seeing things within a group of professionals, in this case fitness professionals, contingent on a particular culture and sets of practices made meaningful within that particular group. Professionalism thus occurs by means of coding, highlighting, and articulating material representations (Goodwin 1994, p. 606). To analyze how social distancing affects group fitness instructors in their work, their relationship to clients, as well as to the fitness industry, we employ a concept coined by Arlie Hochschild (1983, p. 15), "emotional labor". Hochschild suggests that "emotional labor occurs when employees introduce or suppress emotions in order to portray themselves in a certain light that, in turn, produces a wanted state of mind in another." According to Hochschild, the process of suppressing emotions and producing imageries (in this case, group fitness activities and how to be an instructor) is often shaped by institutions (e.g., Les Mills International) or other social structures (e.g., healthism) (Brook 2009). For example, it can be argued that group fitness instructors have been "consumed" or shaped by LMI's trainings and handbooks, through which they are taught how to think about and teach standardized classes, with the goal of turning fitness clients into regulars. LMI instructors are also encouraged to socialize with other instructors and trainers in the LMI community, in order to become and feel like a part of the global enterprise. Clearly, however, the pandemic has interrupted learned and familiar behaviors representing how to be an "authentic" instructor face-to-face, dividing the fitness instructor community into camps of individuals with differing views on how to maintain a professional identity (through emotional labor and rituals), which is largely based on sociability, while simultaneously maintaining social distancing. The most common narrated way of continuing the profession as a LMI instructor through the crisis has been to give classes online. To gain a deeper understanding of this transition, we consider Randall Collins (2004) concept of *interactional ritual chains*, to reason around why instructors might feel distressed by digital teaching, arguably, since the usual components of a successful ritual (co-presence, mutual focus of attention, rhythmic entrainment (Collins 2020, p. 478)) are not necessarily successfully working via technical devices. Group fitness classes could be argued to consist of pre-structured rituals that are re-lived through each class—these rituals contribute toward group belonging and increased emotional energy (Collins 2020, p. 480). However, during times of physical distancing, Collins (2020, p. 489) argues that virtual socializing has led to "failed interactional rituals", causing, e.g., so-called zoom fatigue as well as emotional fatigue. Collins' concept of rituals is closely related to Charles Goodwin

---

2    Preliminary results from an online survey in Flanders (Scheerder et al. 2020) suggest that people consider themselves to have more time to move since corona/lockdown. Of the already actively sporting respondents, 36% reported moving more, 23% less, and 41% the same. Noticeably, the ones who were not very active (less than once a week) (58%) move more (in contrast: 30% do not move at all). The biggest obstacles preventing physical exercise named are sport infrastructure closed (50%), no friends to train with (30%), and fear of infection when doing sports (12%).

(1994) term *professional vision* that he defines as "socially organized ways of seeing and understanding events that are answerable to the distinctive interests of a particular social group" (p. 606). Whereas Collins focuses on the recreation of acts (rituals) and their felt and manifested outcomes in relation to group belonging, Goodwin meticulously investigates both which types of rituals, cultural materials, and discourses constitute a group of professionals in the first place, while, similar to Collins, he underlines that "all vision is perspectival" (p. 606). Accordingly, to move beyond the term "zoom-fatigue", these analytical concepts aid our discussion in exploring why participants feel both dissatisfied and emotionally drained by virtual fitness.

## 4. Research Design

This study was initiated by the authors in early 2020 at the outbreak of COVID-19. Using qualitative measures and a case-study-based approach, we conducted focused group discussions with LMI-certified group fitness instructors. The informants in this study had previously participated in an anonymous survey called "Being a LMI instructor" that investigated motivations behind the fitness profession (Andersson and Vogl 2019). The survey initially recruited participants by means of a snowball sample launched both via social media channels and through LMI's own newsletter. At the end of the questionnaire, participants had the possibility to provide their contact information if they wished to partake in a follow-up talk. The investigators then contacted individual respondents based on their geographical location, gender, and certificates—the investigators desired an even gender quota, as well as inclusion of instructors with different LMI certificates (instructors, advanced instructors, presenters, national trainers).

The advantage of using a case study is that it makes it possible to achieve a rich and nuanced portrait of what might be considered a rare case (Yin 2014). In this sense, this paper can be read as an investigation of group fitness professionals' ways of dealing with a particularly extraordinary situation (a pandemic), and the impact this has on their occupation. Its contribution thus lies in connecting the subjective—the experiences of the professionals—to transformations in perceptions of health and fitness in relation to a historical event currently impacting contemporary society.

Data for the study were gathered in two sets. In April 2020, four focus-group interviews, with a total of ten participants in clusters of two to four people, were conducted. The participants were active LMI instructors from nine countries: Austria, Belgium, Finland, Israel, Sweden, Spain, Germany, Jordan, and the US. Participants were invited to participate on a voluntary basis after having filled out an online questionnaire concerning their fitness careers. The interviews ranged between sixty and ninety minutes in length and were audio recorded. Among the participants, nine respondents identified as women and five as men. Ages ranged from twenty-seven to forty-eight. Due to the international compositions of the focus groups as well as to restrictions concerning social distancing, the group interviews were conducted online, via Skype and Zoom. Using focused group discussions made it possible for participants to engage in processes of "sharing and comparing," which ideally "provides insights into both what participants think and why they think the way they do" (Morgan and Kim 2018, p. 250). Similar as to how it has been described that a pandemic moves in waves, one can imagine that attitudes are also dynamic and change over time. To best capture such changes, we deemed it productive to communicate with the participants over time. In September 2020, the second set of data was gathered through three focused group discussions involving the same participants. The interviews were thematically organized and semi-structured (Atkinson 2015). The themes covered participants' narratives of becoming fitness professionals, their current national situations/directives, their ability to practice their profession, their thoughts about the development of online on-demand services for fitness clients, and their experiences of social distancing.

Furthermore, in the sampling, differences in socio-economic status (e.g., LMI full-time instructors vs. instructors whose main job was outside the fitness industry), living situation (living alone or with other family members or friends; with or without children), and

national context were considered beneficial for the ongoing discussions as lockdowns were implemented differently depending on each country's rules, varying from, for example, governmental recommendations and partial travel restrictions in Finland to a virtual curfew in Israel and Jordan. Including informants from various countries, indeed, shows how different measurements and restrictions can cause differing perceptions of COVID-19 and its severity. For example, in September 2020, LMI instructors participating in the study from Scandinavia and Austria, where gyms were open, expressed feeling "back to normal", whereas instructors from the USA, who could only exercise with face masks, rather felt as being "in the midst of the crisis" (Vogl et al. 2021).

In our presentation of the findings, we have chosen to use a descriptive approach. This was done with the intention of creating depth in our understanding of how COVID-19 has impacted the (professional) lives of the participants. We had no desire to separate the data presented from the analytical framework presented; instead, we considered the experiences and narratives presented as being theoretically impregnated (Tavory and Timmermans 2009; Gomm 2000). Since this study concerns the professional roles of instructors, the survey did not need to go through an ethic clearance. This fact, however, does not give us the right to use data as we please. Some group interviews may, for example, include sensitive discussions. Based on this, we limited our analysis to focus on excerpts that facilitated relevant analysis and refrained from using sensitive or personal information.

## 5. Findings

### 5.1. "It Is Called Group Fitness for a Reason"—Social Distancing and Social Connectedness

In times of social distancing, sport activities that can be performed outdoors clearly have some advantages. Contemporary group fitness activities, however, are usually performed indoors. While it can be argued that it would be possible to move group fitness outdoors, by minimal means, such attempts have been limited. Instead, online streaming has been the most common alternative way of delivering fitness instruction, adhering to social distancing. In alignment with both Collins' and Goodwin's understanding of social scripts and professionalism, prior to the pandemic, giving a class rested on a well-rehearsed script of discursive practices that were reproduced during each session, and, therefore, many informants expressed reluctance toward even trying to teach online. Among the informants, only a few had themselves taught online classes, although most had made use of on-demand fitness services—the general attitude toward streaming classes was negative.

Pekka, 49 years old, from Finland, is a full-time fitness entrepreneur and instructs various streamed classes from a room in his private home. He described the new trend as "better than nothing" but pointed out that it feels unnatural to talk into the camera as opposed to direct interaction, and that his room is too small for group fitness. He explained that every time he kicks or makes a certain move he is thinking, "where is the bookshelf." Another participant, Joy, a Brazilian woman in her thirties working full-time in a gym in Washington, D.C., also talked about the difficulties that come with the new conditions for teaching group fitness classes. Her gym closed completely during lockdown and she was one of the few instructors who taught a lot of streamed classes to gym members. She stated that:

> . . . to bring enough energy that it reaches into someone else's living room is really difficult and afterwards you turn the lights off and you're by yourself ( . . . ) it is much more emotionally draining to do it that way because there's nobody there.

The general consensus among the participants was that teaching live streamed classes in times of social distancing is an unattractive option, because the social aspect and the connectedness with others are missing—an indication that some social rituals fail through the use of screening options (Collins 2020). In the words of Klaus, a photographer and part-time instructor based in Vienna:

> It's not for me. I do this because I enjoy the classes, not because I have to do them. I enjoy them because there are people participating in the classes. That's

what creates the energy, and that's what's fun about the whole thing, and you
have none of that if you're doing live-streamed, so that's an absolute last resort—
perhaps you could save it for just before the apocalypse (laughs). But I think
I would even rather go to a park and do the workout alone than teach a live-
streamed class.

Klaus explained that while the gym that he usually frequents and works in was closed, he
did not exercise. He found it challenging to motivate himself when the social dimension of
his work changed, and so did the socio-emotional "payback." Teaching classes was (and is)
for him first and foremost a sociable hobby that produces emotional energy.

In regard to traditional group fitness activities, Pekka was the only informant who
taught face-to-face classes during lockdown. He said that classes were limited to nine
participants, excluding the instructor. He also stated that the schedule was strongly reduced,
which affected his level of income. He explained that in times of social distancing, "people
understand what group fitness means to them as a hobby and in their lives." However,
Pekka was also aware that it could be seen as controversial to teach face-to-face classes
and was therefore not posting any content of his classes on his social media accounts. He
recalled that he was once severely criticized online for not staying at home. Thus, what
this situation shows is that although certain measures were taken to deal with or avoid any
spread of the COVID-19 virus, Pekka still had to balance his aim of continuing to practice
his profession with others' conceptions of how social distancing should normatively be
upheld in public discourse. Additionally, although classes were running, a Swedish
instructor added that trainers at her gym were not even allowed to promote their classes
on social media.

According to Hochschild (1983, p. 85) argument, "in the public world of work, it is
often part of an individual's job to accept uneven exchanges, to be treated with disrespect
or anger by a client, all the while closeting into fantasy the anger one would like to respond
with." Correspondingly, Pekka, who relies on his income as an instructor, had to pay
attention to his professional image and perform emotional labor in order to uphold his
professional identity while simultaneously preserving the social dimension of group fitness
activities. Another participant, Sam, who is an English instructor currently residing in
Barcelona, decided to teach some pro bono outdoor face-to-face classes at a basketball court,
although it went against the wishes of his gym. He, too, touched on the social dimensions
of his work and how it relates to his emotional state. In his words, "I missed the endorphin
kicks I usually would get two or three times a week, and it was really challenging, since
it used to be such a big part of my life." Arguably, Sam and Klaus were not prepared to
invest as much emotional labor as Pekka, as they were not financially dependent on their
fitness classes.

Although each instructor expressed some degree of disappointment in online teaching,
Joys' narration most clearly depicts a "failed ritual" as she speaks of feeling emotionally
drained from teaching online classes. The fatigue that she portrays seems to indicate a lack
of positive feedback processes that otherwise occur when she sees her participants face-
to-face doing "highly coordinated social interaction" (Collins 2020, p. 480) that otherwise
results in a feeling of rhythmic entrainment (increase in mutual excitement often as a result
of dancing or singing (Collins 2004)). She reports that she uses a chat function to simulate
an accustomed class, beginning with small talk, asking people questions, arguably, to create
mutual focus of attention and shared emotions (Collins 2004). Nevertheless, Joy confesses,
"I do it mostly to motivate myself".

Countries employed various strategies when opening up after lockdowns and in
some cases closing down again. In Austria, for example, fitness studios slowly began
to re-open after what was later considered the first wave at the end of May 2020, and
by September, classes were running almost normally again until the second lockdown
began in October 2020. In the USA, as two of the participants (Kayla and Joy) explained,
instructors could only teach classes wearing masks. This was initially difficult, but they
both said they have adjusted well. Most of the other participants would not consider

wearing a mask to teach, and Jamal even said that "teaching a class while wearing a mask is more dangerous than corona." Due to the risk of contracting the virus at a gym or fitness facility, there were also some participants that were hesitant to return to their gyms. During our conversations, it became apparent that participants found it difficult to be "good" instructors while complying with governmental restrictions. This could be interpreted as a balancing act between being a global LMI instructor who ought to promote fitness participation positively at all times, and a local fitness employee that must also adhere to local restrictions in a gym. For example, Linda reported that since her gym is only allowing ten participants per class at the moment, she leaves the classroom door open in order for clients who did not sign up fast enough for a spot to participate from outside the room.

> "I don't know exactly what to tell them. I guess it's ok as long as they are not inside. It's difficult to turn people away."

Linda's narrative illustrates conflicting imperatives—adhering to restrictions versus a desire to be inclusive—an otherwise paramount trait of instructorhood.

In summary, all of the respondents, apart from Pekka and Joy, were reluctant to teach online, since they felt that group fitness should be about coming together physically and socially, manifesting a strong wish to maintain accustomed rituals rather than adapting new ones. Various moral stances toward social distancing and how to perform it contributed to an emotional distancing among instructors and their clients. Navigating the professional role and image in relation to both health recommendations in different countries and clients' demands involved significant emotional labor that expressed itself in various ways. For example, Jamal considers wearing a mask while working out to be dangerous whereas Kayla and Joy consider it a "necessary evil," required to fulfil the professional tasks placed on a group fitness instructor at a gym. Importantly, since the first round of discussions, some instructors also changed their attitudes toward using masks during classes, beginning as something that would be "an absolute last resort" (Klaus, Vienna) to an everyday practice.

Additionally, read through Goodwin (1994) argument, the physical space of the gym emerges as a "key locus for their practices (teaching), the place where nature (movement to music) is transformed into culture (a class)" (p. 608). For instance, one part of their professionality is shown through their musicality—an element that is potentially threatened when using streaming devices where audio and visual elements might lag or work conditionally only—making it impossible to display professional literacy (Goodwin 1994, p. 612).

*5.2. Strategies for a Strong Instructor Community: #LesMillsUnited*

LMI refers to itself as a global family whose members are united by their passion for movement. They distribute the same standardized classes to their instructors all over the world, and instructors can even buy LMI-branded clothing that shows what programs they teach and, possibly, also the lifestyles they "sell." To ensure that instructors have opportunities to get to know other instructors, LMI usually organizes various mega-events on a yearly basis. Here, instructors can purchase tickets to exercise with thousands of other instructors simultaneously, while being taught by well-known LMI "celebrities." Thus, one can argue that, although LMI instructors have agency, "the parameters of their work have been established by the system that is organizing their (professional) perception" (Goodwin 1994, p. 609).

Even though LMI continued to produce workouts during the 2020 pandemic, all face-to-face social events among instructors were cancelled. Therefore, to stay in touch, many instructors were highly active on social media throughout lockdowns. Different LMI programs also have official Facebook groups, and most gyms connect their instructors via WhatsApp or other direct messaging services. To keep instructors motivated at home, LMI launched the #LesMillsUnited campaign on the 1st of April 2020, in which they encouraged people to film short clips of themselves and upload them onto their social media accounts, using the corresponding hashtag. Subsequently, they filmed their teaching materials as

a "united release" in which instructors could be seen performing the workouts at home or alone in various outdoor locations—illustrating togetherness while practicing social distancing. The campaign was supposed to be a "positive vibes only" endeavor—terms such as corona, COVID-19, lockdown, etc., were discouraged.

The respondents in this study were asked about their opinions concerning the campaign in both April and September, paying special attention to whether the campaign strengthened their sense of belonging to the brand and the trainer community. The responses generally indicated indifference, but some were partly negative. Three of the respondents, however, Kayla, Jamal, and Pekka had favorable views, perhaps not surprisingly, since they had been able to personally participate in the campaign. Jamal explained,

> it represents all the values that LMI stands for. They say we are going to change the world, and it's nice to see people from all over. In LM Middle East, we had one move each day for 30 days and it worked out really well.

Jamal appreciated LMI's goals and confessed to find their workouts addicting. He considered the difficult coronavirus phase as a time in which instructors should live in accordance with LMI's values and be fitness role models. Kayla, based in the USA, who also actively participated in the campaign as a presenter in a filmed step-cardiovascular class (Bodystep™), said that it connected her with many other international instructors, since she was invited into various new social media groups. Pekka, who was the only instructor who actually taught the routine to participants face-to-face, reports that the release was received positively within his "fitness flock." Two other participants said that the campaign "shows off how big the community is". Klaus, on the other hand, explained,

> To be completely honest, I think the campaign is quite depressing [laughs]. They're trying very hard and it's of course better than nothing, but it's definitely not what it used to be, or what it's supposed to be, in my opinion.

Klaus referred to the fact that classes without participants and a stage feel boring, whereas Jamal felt empowered by seeing instructors exercising in other geographical locations. Ultimately, Klaus primarily considered LMI a fitness provider and felt disappointed by the quality of the workout, while Jamal stressed a sense of community seemingly unrelated to the standard of the material. Nevertheless, most of the answers conveyed an indifferent stance—for example, Anita, who confessed that she did not actively participate by uploading any footage,

> I think that they did their best in a very strange situation, trying to engage instructors by putting together different clips so it looks like people are training together in this very strange situation in which we cannot actually be together.

Clearly, Anita also interpreted the campaign as a team-building attempt but simultaneously agreed with Diana who states, "it did nothing for me." The indifference toward the campaign might tell us something about the value of the workouts to the instructors; although it is a global community, the acquainted participants and personal colleagues in the local studios appear to be more important. Interactions with befriended fitness enthusiasts seemingly enables successful interactional ritual chains, which, in turn, generate belonging and socio-emotional connectedness. A further cause as to why participants might not have felt engaged in the campaign is that when they finally received the didactic material from the #LesMillsUnited release, some were no longer in lockdown. Both Klaus and Linda agreed that "it would've needed to be launched three months earlier." Additionally, some participants said they were disappointed that LMI did not lower the costs of the workouts that the instructors have to purchase, even though, as they would have known, most of the instructors were unable to teach classes. In the end, the consensus appears to be that the campaign could be considered a nice gesture but that it did not necessarily achieve the "unitedness" or social connectedness that it set out to accomplish.

### 5.3. Post-Corona: The Future of Group Fitness

Physical exercise has been taken for granted as a personal prerogative, and yet with restrictions of movement and strict curfews, many have had to rethink their time outdoors and performance of sport activities (Andersson et al. 2021a, p. 3). As a consequence, there have been strong reactions against restrictions of movement; for instance, Nike's current campaign, "You can't stop us" communicates that an active lifestyle is firmly grounded in a democratic society. Several informants expressed dissatisfaction with the closure of gyms, in alignment with the #touchepasmasalle campaign that originated in France as a reaction toward closed fitness facilities (FranceActive 2020). The campaign, in which LMI also participated, urged gyms to be allowed to reopen with the motivation that gyms are "a part of the solution, not the problem."

Still, unable to control whether gyms are closed or not, most informants said they used the imposed break to educate themselves. Looking forward to a near future, Melissa asserts that,

> This is a perfect time to reinvent yourself as an instructor. Ask yourself how you wanna come back. I think that we will develop new teaching skills through the virtual classes in terms of how to keep people interested.

What Melissa and others here exemplify is the entrepreneurial spirit and inventiveness that encourages using new, or at least supplementing existing rituals, that have been characteristic of the fitness revolution of this century (Andreasson and Johansson 2014). In fact, the gym and fitness industry has been prominent in its efforts to combine and incorporate different training elements (e.g., stretching, martial arts, cycling and weightlifting), creating new hybrid and complex training opportunities for consumers (Sassatelli 2000). This inventiveness has also been promoted to and by fitness instructors, and is used as a cultural script by which to approach plans for the near future. One participant residing in North Carolina, Glen, correspondingly felt that the current circumstances provide opportunities to grow and develop as an instructor. As he recollected, "every time I teach another workout from at home, I realize how I can improve some small detail". Jamal relates that the closure of gyms has taught him that he needs to learn another profession in case of another crisis. Simultaneously, he is positive about the future and his and others' ability to take on challenges yet to come, seemingly based on essentialist ideas of human behavior:

> It forced me to think about other things I can do besides fitness, because who knows when something like this can happen again, so we need to have a back-up plan, and it should be something completely different from what we do. / . . . / Humans throughout history have always endured—we are gonna find solutions. I don't think that live group fitness classes will be completely cancelled either now or in the future. I think humans are creatures of connection. That's what makes us human, so we will adapt and find solutions. (Jamal)

Jamal clearly underlines that the pandemic has made him realize that one must be willing to think outside of the box to remain successful and independent, according to him, something that humans have always managed throughout history. Although not all informants wish to find alternative professions, most do seem to interpret the imposed time-out (Krieger et al. 2021) as an opportunity to improve themselves in their current lines of work. This aligns with an entrepreneurial zeitgeist that portrays personal development as morally good and necessary. As argued by Brown (2012, p. 54) "people must tend to their current and future value on the market." For example, as Pekka reported, "I decided I was not gonna lie on my couch and feel sorry for the world, this was my time to be better." The urge to develop further in a linear upward-like manner could help instructors to maintain a positive mind-set during the crisis, but it could also put unnecessary pressure on instructors to perform emotional labor.

In countries like Sweden and Finland, where recommended safety measures have been voluntary, some report that they felt torn when deciding on the extent of social distancing that they should perform. One respondent reports that he even experienced "Net hate"

after teaching face-to-face classes, and Ylva, a Swedish instructor, points out that she never before had to justify why she gives a class, presumably since most have considered fitness as an acceptable health-promoting practice. However, during the pandemic, her gym in a smaller city was under attack from upset citizens, demanding they close the gym to prevent the spread of COVID-19—a development she considers troubling also when looking into the future of fitness.

On a structural scale, it is meaningful to consider how group fitness might be affected in the long term by the necessity of social distancing in the wake of a pandemic. Although, e.g., Jamal thinks that online classes might be a larger part of group fitness in the future, most other participants disagree. When interviewed in September, as the infection rate in Sweden was decreasing, Anita could see how the number of gym-goers at her gym was increasing, getting back to "normal", although in a slightly new form.

> I don't think that the gym was ever as clean as it is now. People feel safe and keep out of the way and show consideration. I think that we are social creatures who need to be together physically. You want the energy and you want to share the experience with people, and you can't do that with live streaming in the same way.

Anita's comment illustrates how following guidelines might increase a feeling of safety and control during the pandemic, a fact she believes will result in customers returning sooner rather than later.

## 6. Discussion and Conclusions

This paper has explored how the first and partly second wave of the corona crisis have affected respondents' everyday lives and professional possibilities. Before the pandemic, which began in early 2020, the participants in this study would probably have found it difficult, or even amusing, to address the idea of a life without LMI classes. Answering such a hypothetical question would have included a likelihood of something that could threaten the existence of a multinational company and their personal ability to pursue their chosen line of work. However, perspectives and attitudes can change quite rapidly—the COVID-19 pandemic has altered both current and possibly future prospects of being an LMI instructor. Like various other professional settings, the fitness world has had to navigate onto digital platforms, replacing face-to-face classes with live streaming and pre-filmed on-demand routines. This relocation, although it allows group fitness to continue, challenges some of the core values of the respondents' professional identity, since it is largely based on social dimensions that require certain rituals that cannot be successfully performed through technical devices. Ultimately, the current restrictions that are forging less personal contact are a trend toward which the respondents appeared skeptical. Importantly, a distinctive aspect of being a LMI trainer is the performance of the workout with the group. Both client and trainer perform the exact same moves for the duration of the whole class, which creates a symbolic bond and rhythmic entrainment of having gone through a shared experience. Hochschild (1983, p. 15) argues that "the worker invests her energies and life in the production of the commodity, it is a one-way relationship—a "one-sided enrichment of the object". However, usually, trainers do not seem to feel that it is a one-sided service, since they are also doing the exercise—taking part in the physical and mental benefits of the training. As a result, instructors may give up the profession—especially those who are working part-time. Although the failed rituals do not seem to cause instructors to quit LMI, informants convey a certain degree of anxiety that their clients may not come back as time passes without classes; some argued that clients who turned to on-demand services during lockdown will want to stay with that alternative, since it is seemingly convenient.

As this study was initiated in April 2020, the idea was to ignite a dialogue among instructors and to follow them throughout the COVID-19 pandemic. The follow-up talks that were held in September 2020 were initially planned as post-corona conversations. However, as some countries once again tightened their restrictions due to the disease, the pandemic was by no means over at this time, and participants questioned if there

will even be a real post-corona era. Especially, due to the longevity of the pandemic, participants came to reflect on exercise—its role in society and who it is intended to benefit. Once group fitness could be practiced again, would everyone, including risk groups, be able to participate? If not, might "protective" restrictions, e.g., a vaccination certificate, become exclusionary[3]?

Participants were explicitly asked what has been most challenging during this period. Nadja, for example, said that the lockdown opened her eyes to how fast the status quo might change within the fitness industry. She lost her trainer position during the pandemic and has not yet been able to return to her classes. Sam, who presently teaches much less than before, said "you wanna think that you're irreplaceable, but they're addicted to the classes, not to you, and there's always gonna be a new instructor to take your place." This comment illustrates how standardized workouts correspond to a franchise system in which the workouts stay the same while staff may be easily replaced.

The #LesMillsUnited campaign, which was supposed to increase the sense of belonging within the trainer community, generally fell flat among the participants. Instead, they reported that keeping in touch with local colleagues online felt empowering. Based on country-specific measurements, the campaign came across differently in different countries. For example, participants in USA, who were wearing masks in their gyms, showed more enthusiasm than respondents in Sweden, Finland, and Austria, who reported that their situation at the gym was "almost back to normal." To remain motivated at home, many, for instance, Pekka, created an action plan to successfully navigate through the crisis with the "right" balance of nutrition, exercise, and workshops—arguably—to stay attractive on the competitive fitness market. The minority of respondents who felt a lack of motivation to exercise at home or practice healthy eating expressed feelings of guilt feeling "lazy." Accordingly, the results indicate that an excess of spare time and increased anxiety have resulted in participants exercising even more than before, which can be seen as both stress-relieving and disciplining (Malcolm and Velija 2020, p. 31). Participants who expressed a need to holistically improve during quarantine seem to conform to healthism, which is rooted in both fear of disease (exercising to stay clear of the COVID-19 virus), and personal failure (failing to provide emotional labor) (Crawford 1980). Therefore, two forces, both of which are related to healthism, seem to be at work simultaneously: living up to an idea of how a fitness trainer "should" look/perform, and improving as a person to be able to be an "authentic" fitness role model.

From a broader perspective, this study allows some insights into how the neoliberally driven fitness industry, as well as individuals constituting this business, have adapted during the pandemic. As the informant Jamal points out, it seems that adaptations, or perhaps redefinitions are taking place, since the fitness industry has long relied on successfully promoting its services as a pursuit of optimizing personal health. Concurrently, moral dilemmas and governmental restrictions related to the pandemic have altered the possibilities to pursue personal health, forcing gyms to rethink their strategies, resulting in arguments that portray gyms as a part of a solution—essential businesses—e.g., the #touchepasmasalle campaign from France. One can detect a distinctive ideological shift from exercising as a "cool lifestyle choice" to being portrayed as an "inherent" need and right of every individual (that, arguably, is best maintained in a gym). Already at this early stage, one can observe how this phenomenon, seemingly successfully, produces a dichotomy of people who are allegedly afraid of COVID-19 (who stay at home), and people who are allegedly not afraid (who continue). The outcome is a creation of a new, neoliberally driven subject position, in which people, despite severe restrictions, can emerge as "strong and determined fitness supporters"—waging a war against the virus by tending to their physiques (Vogl et al. 2021). For instance, the interviewee Pekka reports that he will not stop visiting the gym unless it becomes illegal to go there, exclaiming, "fuck corona."

---

3    For example, in March 2021, Israel began opening up gyms to people who can prove that they have been vaccinated. In a similar vein, both local and global political discussions are concerning applying similar procedures to enable visiting theatres, traveling, and gastronomy, which could rapidly bring forward class-related issues—those who will be granted access to the vaccine early on will possibly have a privileged position in society.

Finally, there are, naturally, limitations to this study; the participants, although residing in different parts of the world, are not representative of the whole LMI community or group fitness in general. However, their perspectives allow for insights into how a particular sub-group of global fitness professionals navigate through a global crisis, and by following the same individuals over time (interviews were held with the same participants in April 2020 and September 2020), one can obtain nuances of diverse manifestations and perspectives on the COVID-19 pandemic. A further important aspect to address is the social component of group fitness—how can it be maintained long-term alongside social distancing? This article has touched upon such dilemmas that were circulating within fitness culture, and although there are no absolute answers to these questions, it will be an important task within the sociology of sport to thoroughly address these topics.

**Author Contributions:** Conceptualization; K.A. methodology; K.A. & J.A. software; n.a., validation; K.A. & J.A., formal analysis; K.A., investigation; K.A., resources; K.A., data curation; K.A., writing—original draft preparation; K.A., writing—review and editing; J.A., visualization n.a., supervision; J.A., project administration; K.A., funding acquisition; n.a. All authors have read and agreed to the published version of the manuscript.

**Funding:** This project did not receive any funding.

**Institutional Review Board Statement:** Ethical review and approval were waived for this study, since the study investigated professional opinions only. Any sensitive information was excluded and interviews were only audio-recorded.

**Informed Consent Statement:** Informed consent was obtained from all subjects involved in the study.

**Data Availability Statement:** Not applicable.

**Conflicts of Interest:** The authors declare no conflict of interest.

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
