# Peer review of "Being a Group Fitness Instructor during the COVID-19 Crisis: Navigating Professional Identity, Social Distancing, and Community"

_socsci, doi:10.3390/socsci10040118_

Round 1

Reviewer 1 Report

The article displays a relevant problem for the fitness industry and its professional dimensions and also for the sociology of sport. It is well-written and clearly structured. Some revisions and it will make a fine contribution to the journal and to its field.

The problem with Covid-19 is obviously original since we are still in the beginning of the expected tsunami of studies of the pandemic that will populate journals for years to come. It is good to be first at the ball, and it is interesting to read about how one globally spread phenomenon (Corona) is affecting another globally spread phenomenon (Les Mills). 

Relevance aside, I believe that the discussion will merit from some more support on the theoretical side. There are some interesting concepts being introduced, that needs to be elevated in the discussion. How does the new insights of these found instances of Healthism relate on a larger scale to neoliberalism and McDonaldization? Could the nationalities of the participants play a role here?

What I found undertheorized is the concept of Professionalism. Profession is a rich concept in sociology, both with respect to its history and to its different definitions. The article needs to make a stand in this regard. Why Profession(alism)? What is meant by profession? What is retained from sociology? Is it about teacher Professionalism, i.e. from a more educational perspective?.

There is one recurring theoretician in the article that talks about emotional labour, which could be a good start. However, for the moment it appears as an underdeveloped theme. Perhaps there is a healthy little lineage to classical sociology here to be made, albeit briefly, namely to Durkheims concept of "collective effervescence" (which to me seem to be the ideal that the Les Mills instructors are missing during the pandemic). Also, Goffmann's impression management might be mentioned in the build-up.

Another Way, decidedly more theoretical, but perhaps more original, would be be to go with Goodwins anthropological view on "Professional Vision" in his seminal article with the same name. In it, an elaborated discussion on  profession-specific views developed, trained and contested by practitioners is offered. That would be really theoretically intricate to apply to their visions of themselves on films, when they apply, on films during the campaign, on social media, their vision of the classes, of health, their shame, and their bodies, their training, their own professionalism, even. It seems as if the crisis of those instructors in this situation is where to now direct their professional vision: their own bodies, cameras, nutrition, society, etc?

What really seemed to resonate with the findings, but in an inverted manner, is the professionalism of gamers broadcasting from Twitch. They interact in highly personalized, creative and ingenious ways with their audiences from their homes. Having no problem at all engaging the ones they interact with (and not worrying about it). Their, the gamers', professionalism from home was never McDonaldized, but highly original. The LesMills instructors came from McDonaldization, and now they face... what? If the authors ever come across the book Watch Me Play (Taylor 2018)(https://press.princeton.edu/books/paperback/9780691183558/watch-me-play), I think that an exciting discussion, other than about zoom fatigue, could be addressed, one that is about our bodies, our connectedness, our impression management, and about professionalism from home.

Minor remarks:

How the data was handled needs to be clarified. I don't really understand the arguments of "separation", etc. 

Layout needs to be worked upon, block quotes must differ from the rest of the text, and hyphens must be removed in the middle of words when they appear in the middle of a line.

Author Response

Thanks for the constructive feedback and critique provided by the editor and the reviewers, which helped us bringing the piece forward. We have tried to meet the comments to the best of our ability and generally had few objections to the suggested revisions. Below we explain in detail how we have addressed each comment. Again, we thank the Editor and Reviewers. 

The article displays a relevant problem for the fitness industry and its professional dimensions and also for the sociology of sport. It is well-written and clearly structured. Some revisions and it will make a fine contribution to the journal and to its field. The problem with Covid-19 is obviously original since we are still in the beginning of the expected tsunami of studies of the pandemic that will populate journals for years to come. It is good to be first at the ball, and it is interesting to read about how one globally spread phenomenon (Corona) is affecting another globally spread phenomenon (Les Mills).

-thanks for the encouragements 

Relevance aside, I believe that the discussion will merit from some more support on the theoretical side. There are some interesting concepts being introduced, that needs to be elevated in the discussion. How does the new insights of these found instances of Healthism relate on a larger scale to neoliberalism and McDonaldization? Could the nationalities of the participants play a role here?

Thanks, this ties in with our discussion concerning the future of group fitness where we point toward how instructors navigate their instructorhood in ways that can still be considered to apply to neoliberal ideals. For instance, online educations, new personal training programs that are specially designed to “make them stronger for after the pandemic—coming back stronger.” But also mindsets such as “waging a war against the virus” through a strong immune system that is narrated to come with a lean physique. We have now added more on this topic into the discussion.

What I found undertheorized is the concept of Professionalism. Profession is a rich concept in sociology, both with respect to its history and to its different definitions. The article needs to make a stand in this regard. Why Profession(alism)? What is meant by profession? What is retained from sociology? Is it about teacher Professionalism, i.e. from a more educational perspective?.

After considering different options we have decided to use Goodwins’ definition of professionalism, since it suits within the LMI concept. Thanks

There is one recurring theoretician in the article that talks about emotional labour, which could be a good start. However, for the moment it appears as an underdeveloped theme. Perhaps there is a healthy little lineage to classical sociology here to be made, albeit briefly, namely to Durkheims concept of "collective effervescence" (which to me seem to be the ideal that the Les Mills instructors are missing during the pandemic). Also, Goffmann's impression management might be mentioned in the build-up.

- We have used Randall Collins (2004) for the analysis, who bases his insights on the mentioned scholars, which is the reason that we have not worked with these per se. Thanks

Another Way, decidedly more theoretical, but perhaps more original, would be be to go with Goodwins anthropological view on "Professional Vision" in his seminal article with the same name. In it, an elaborated discussion on  profession-specific views developed, trained and contested by practitioners is offered. That would be really theoretically intricate to apply to their visions of themselves on films, when they apply, on films during the campaign, on social media, their vision of the classes, of health, their shame, and their bodies, their training, their own professionalism, even. It seems as if the crisis of those instructors in this situation is where to now direct their professional vision: their own bodies, cameras, nutrition, society, etc?

- We have integrated Goodwins’ terminology as a way to build upon Randall Collins who theorizes around a similar idea but focuses “only” on the rituals. We think that a combination has widened the discussion on digitalization to move beyond the idea of “zoom fatigue.”

What really seemed to resonate with the findings, but in an inverted manner, is the professionalism of gamers broadcasting from Twitch. They interact in highly personalized, creative and ingenious ways with their audiences from their homes. Having no problem at all engaging the ones they interact with (and not worrying about it). Their, the gamers', professionalism from home was never McDonaldized, but highly original. The LesMills instructors came from McDonaldization, and now they face... what? If the authors ever come across the book Watch Me Play (Taylor 2018)(https://press.princeton.edu/books/paperback/9780691183558/watch-me-play), I think that an exciting discussion, other than about zoom fatigue, could be addressed, one that is about our bodies, our connectedness, our impression management, and about professionalism from home.

- Thanks, it is an interesting point that some communities seem to rely on exactly that which the LMI community does not wish to apply. There are, however, some distinct differences between the Twitch community and LMI, also relevant when speaking of professionalism. For example, Twitch users are not standardized or usually earning money when they are online. Additionally, since their joint activity is gaming, their venue has not changed due to the pandemic, but rather boomed since people (arguably) spend more time at home.

Minor remarks:

How the data was handled needs to be clarified. I don't really understand the arguments of "separation", etc.

- Thanks, we have added some additional information into the methodology section.

Layout needs to be worked upon, block quotes must differ from the rest of the text, and hyphens must be removed in the middle of words when they appear in the middle of a line.

-We have looked over the layout, hopefully it reads better now.

Reviewer 2 Report

I Would like to thank the editors and the authors for the opportunity of reviewing this manuscript. I think the topic is interesting as provide a good insight into what fitness instructors might currently feel and more precisely LMS instructors. However, I think some several limitations and weaknesses should be addressed before considering this manuscript for publication. I hope these comments help to improve the quality of the manuscript.

General comments:

  • Please, review the whole manuscript and correct all words broken into two that are not at the end of a line. It is very annoying reading a manuscript and find this typo every three or four lines. Ej. Page 1. Line 11: “Pandem-ic”; or Line 12 “Profes-sions”

  • I do not think there is a need to leave this amount of space between paragraphs.

Abstract:

  • I think there is no need to describe here what Les Mills is (Lines 13-15). I would recommend saving these words for providing further information in other abstract sections.

  • I think research methods can be improved and provide much further information.

  • Implication: I think the authors should make an extraeffort here.

Keywords: It might be appropriate not to use words that have been already used in the title

Introduction

  • Paragraph 1 and 2: I think that more references might be needed. Mainly in paragraph 2 where the authors state the impact of COVID19 in different countries (page 2 lines 56-61). But also, in other parts of these paragraphs.
  • Page 2 line 67: I think there is a typo here “have all have been…”
  • Line 76: I understand what “(Ibid)” means and how to use it, but I am not sure if it is appropriate to use it here instead of providing the full reference despite it is the same reference as the one used before.
  • I know this is a qualitative paper, but I think it should contain a subheading in methodology in which the characteristics of participants are presented. Instead of providing information on this regard at the end of the introduction.

Background

  • I think the authors mislead the term “sedentary behaviour” with “physical inactivity” which are not the same. The first one is al waking activity conducted in a lying, sitting or reclined position that requires approximately 1.5METs or less. The other one is when a person is not meeting the international guidelines for physical activity. Both are very related but different. So, the pandemic and lockdowns have probably two impacts on people’s life: an increase in sedentary activities and a decrease in physical activity levels with a higher proportion of the population not meeting the international guidelines. I invite the author to check the first and second paragraphs and clearly differentiate among both behaviours.

  • I think this section can be improved. For instance, the authors do not discuss the fact of LMS is most of the time offered in gyms, fitness centres and similar which many of them are not or have not been allowed to open. Also, I think it might be important to analyse the COVID19 risk associated to exercise in these centres. I think it would help to draw the current situation of LMS instructors

analytical framework

  • Last paragraph. I do not understand what this paragraph adds to the analytical framework or to the manuscript.

Research Design

  • I still think that participants should be a subheading. It might be within this section or as a new section.

  • Was the project examined and approved by an external ethical committee? I think this is important even more considering focus groups were recorded. Were they video-recorded?

  • I think that further information on who was recruited for this study is required. Were invited to participate all instructors who completed the survey “Being LMI instructors”? were the candidates selected randomly? How many people were invited to participate? How was selection done? How many people belong to each participant country? Why other countries are not included in this study? There were inclusion and exclusion criteria? Etc.

  • I read the introduction, background and analytical framework, but do not understand why two sets of focus-groups with the same participants were conducted. Were the same questions discussed in both sets? I think it might be appropriate to add a Table or supplement with the questions or the sentences used by the authors to introduce a topic. I think it is important to provide enough information to allow someone else to replicate the study and this is not the case.

Findings

  • Line 390: considering that the virus incidence increases and decrease as a wave I do not think this sentence is quite accurate. In fact, the third wave in European Countries was in many cases stronger than the previous two

  • I think that the way findings are described are not appropriate. I think that the three sections should be in line with each research question. I invite the authors to reanalyse the findings and organised in a way the help to respond to these three questions.

Discussion

  • I think the discussion is too vague and the authors mostly repeat the same comments from the results. I miss comparisons with previous research. I think the authors should rewrite this section.

  • I also miss a paragraph with the main study limitations.

  • I miss further comments related to the selected approach and the framework used to analyse the outcomes.

Author Response

Thanks for the constructive feedback and critique provided by the editor and the reviewers, which helped us bringing the piece forward. We have tried to meet the comments to the best of our ability and generally had few objections to the suggested revisions. Below we explain in detail how we have addressed each comment. Again, we thank the Editor and Reviewers. 

  • Please, review the whole manuscript and correct all words broken into two that are not at the end of a line. It is very annoying reading a manuscript and find this typo every three or four lines. Page 1. Line 11: “Pandem-ic”; or Line 12 “Profes-sions”

-thanks we have revised accordingly.

  • I do not think there is a need to leave this amount of space between paragraphs.

-thanks, we are following the suggested template by the journal, but seemingly there was some alterations to it following the process of submitting the ms. Hopefully it works better this time though.

Abstract:

  • I think there is no need to describe here what Les Mills is (Lines 13-15). I would recommend saving these words for providing further information in other abstract sections.

We were asked by the editor to include this information in the abstract so we will let them decide if this should be changed or not.

  • I think research methods can be improved and provide much further information.
  • We added some information on this, thanks
  • Implication: I think the authors should make an extraeffort here.

-Thanks. We added a short discussion on implications in the paper, but did not manage to fit it into the short abstract.

Keywords: It might be appropriate not to use words that have been already used in the title

 - As keywords and title constitute different platforms for the papers searchability we have decided to have some overlaps here. We added some keywords though   

Introduction

  • Paragraph 1 and 2: I think that more references might be needed. Mainly in paragraph 2 where the authors state the impact of COVID19 in different countries (page 2 lines 56-61). But also, in other parts of these paragraphs.

-we added some references. We have also revised the literature review and updated it.

Page 2 line 67: I think there is a typo here “have all have been…”

-revised

  • Line 76:I understand what “(Ibid)” means and how to use it, but I am not sure if it is appropriate to use it here instead of providing the full reference despite it is the same reference as the one used before.

-revised

  • I know this is a qualitative paper, but I think it should contain a subheading in methodology in which the characteristics of participants are presented. Instead of providing information on this regard at the end of the introduction.

-As the paper utilised a qualitative approach to data we have decided to give some information on this in the introduction just to give the reader a clear enough view of they type and texture of the study. This is then addressed more thoroughly in the section on Research design. Thanks.

Background

  • I think the authors mislead the term “sedentary behaviour” with “physical inactivity” which are not the same. The first one is al waking activity conducted in a lying, sitting or reclined position that requires approximately 1.5METs or less. The other one is when a person is not meeting the international guidelines for physical activity. Both are very related but different. So, the pandemic and lockdowns have probably two impacts on people’s life: an increase in sedentary activities and a decrease in physical activity levels with a higher proportion of the population not meeting the international guidelines. I invite the author to check the first and second paragraphs and clearly differentiate among both behaviours.

- We have decided for one clear definition of PA that we have written out in a footnote upon the first mention in the text. To avoid misunderstandings, we have removed sedentary completely, since we used the terms interchangeably. We hope that this will ensure coherence. 

  • I think this section can be improved. For instance, the authors do not discuss the fact of LMS is most of the time offered in gyms, fitness centres and similar which many of them are not or have not been allowed to open. Also, I think it might be important to analyse the COVID19 risk associated to exercise in these centres. I think it would help to draw the current situation of LMS instructors

- Thanks. Due to a lack of space, since many of the interviewed instructors are working in many different studios as freelancing trainers, we have limited the information to if the gym is open and what precautions they take there.

analytical framework

  • Last paragraph. I do not understand what this paragraph adds to the analytical framework or to the manuscript.

- We have now relocated this section into the background section instead—where it seems to make more sense. 

Research Design

  • I still think that participants should be a subheading. It might be within this section or as a new section.

- Due to a lack of space, we have integrated information on the informants into the text wherever necessary.

  • Was the project examined and approved by an external ethical committee? I think this is important even more considering focus groups were recorded. Were they video-recorded.

Thanks, for pointing out this valid argument. Since this study is conducted outside of Sweden and concerns the professional lives, we have not needed ethic approval. We have now added this information, as well as that we only recorded audio for the purpose of transcribing the interviews later.

  • I think that further information on who was recruited for this study is required. Were invited to participate all instructors who completed the survey “Being LMI instructors”? were the candidates selected randomly? How many people were invited to participate? How was selection done? How many people belong to each participant country? Why other countries are not included in this study? There were inclusion and exclusion criteria? Etc.

- Thanks, we have added comments on how different locations seem to form the perception of the crisis (measurements, restrictions seem to create the reality of the disease). We have described in more detail which criteria we considered in the selection process of informants.

  • I read the introduction, background and analytical framework, but do not understand why two sets of focus-groups with the same participants were conducted. Were the same questions discussed in both sets? I think it might be appropriate to add a Table or supplement with the questions or the sentences used by the authors to introduce a topic. I think it is important to provide enough information to allow someone else to replicate the study and this is not the case.

- Thanks, similar as to how it has been described that the pandemic moves in waves, one can imagine that attitudes of the informants also are dynamic and change over time. To best capture these changes, we deem it productive to communicate with the same informants on several occasions. We have worked with an relatively open and thematicially organised interview guide focusing on the participants understanding of their job in relation to the current situation with the pandemic.

Findings

  • Line 390: considering that the virus incidence increases and decrease as a wave I do not think this sentence is quite accurate. In fact, the third wave in European Countries was in many cases stronger than the previous two

-Good point, revised.

  • I think that the way findings are described are not appropriate. I think that the three sections should be in line with each research question. I invite the authors to reanalyse the findings and organised in a way the help to respond to these three questions.

- Thanks, we have revised the RQ slightly to make a clearer connection to the sub-headlines within the findings section.

Discussion

  • I think the discussion is too vague and the authors mostly repeat the same comments from the results. I miss comparisons with previous research. I think the authors should rewrite this section.

- We have made an effort to connect the healthism debate to the covid-19 situation and to other current research.

  • I also miss a paragraph with the main study limitations.

We have included a paragraph commenting on the limitations of the study into our conclusion.

  • I miss further comments related to the selected approach and the framework used to analyse the outcomes.

- We added a short note on this too. Thanks.

Reviewer 3 Report

First of all, congratulations on the work done. However, I consider a series of modifications and changes necessary to be able to publish it. I think the bibliography should contain more articles indexed in scopus databases; they can be found, for example, in Web Science. I also consider it necessary to review the citations throughout the text so that they follow the rules established by the magazine. I also think it would be necessary to provide more relevant research in the conclusions and discussions section. Also, I think the format of the article should be revised in some sections there is a wide spacing.

Author Response

Thanks for the constructive feedback and critique provided by the editor and the reviewers, which helped us bringing the piece forward. We have tried to meet the comments to the best of our ability and generally had few objections to the suggested revisions. Below we explain in detail how we have addressed each comment. Again, we thank the Editor and Reviewers. 

First of all, congratulations on the work done. However, I consider a series of modifications and changes necessary to be able to publish it. I think the bibliography should contain more articles indexed in scopus databases; they can be found, for example, in Web Science. 

-Thanks, we added some sources for this purpose.

I also consider it necessary to review the citations throughout the text so that they follow the rules established by the magazine. 

-ok, thanks, we have revised according to the style sheet.

I also think it would be necessary to provide more relevant research in the conclusions and discussions section. 

  • We added some references to more clearly situate our result in relation to the scholarly debate. Thanks.

Also, I think the format of the article should be revised in some sections there is a wide spacing.

-revised. Thanks, we have revised according to the stylesheet.

Round 2

Reviewer 2 Report

I would like to congratulate the authors for addressing all the comments I had in the first revision. I have no further comments to add in order to improve the quality of this manuscript.